# Accessing Ancient Population Lifeways through the Study of Gastrointestinal Parasites: Paleoparasitology

Matthieu Le Bailly [1,*], Céline Maicher [1,2], Kévin Roche [1,3] and Benjamin Dufour [1]

1   CNRS UMR 6249 Chrono-Environment, University of Bourgogne Franche-Comte, 16 Route de Gray, 25 030 Besancon, France; celine.maicher@gmail.com (C.M.); kevin.roche.paleo@gmail.com (K.R.); benjamin.dufour@univ-fcomte.fr (B.D.)
2   MSHE Claude-Nicolas Ledoux, UAR 3124, CNRS-University of Bourgogne Franche-Comte, 1 Rue Charles Nodier, 25 000 Besancon, France
3   CNRS UMR 5607 Ausonius, University of Bordeaux Montaigne, 8 Esplanade des Antilles, 33 600 Pessac, France
*   Correspondence: matthieu.lebailly@univ-fcomte.fr; Tel.: +33-(0)381-665-725

**Abstract:** Paleoparasitology is a discipline of bioarchaeology that studies human and animal parasites and their evolution through time. It is at the frontier between biological sciences and the humanities, and aims to provide valuable clues about the lifestyles of former populations. Through examples chosen among recent case studies, we show in this review how paleoparasitology contributes to issues related to food, health, hygiene, organic waste management, and site occupation by ancient populations, but also, in the longer term, to questions of the evolution of the human/animal relationship and the history of diseases. This article provides an overview of this research field, its history, its concepts, and in particular, its applications in archaeology and the history of diseases.

**Keywords:** paleoparasitology; paleoecology; ancient parasites; human; animals; evolution





## 1. Introduction

Paleoparasitology aims to study the natural history of parasitic organisms through the recovery of their preserved remains in archaeological, paleontological, paleoecological, and medical contexts. Consequently, this approach is highly integrative, overlapping different but consistent scientific fields, such as archaeological sciences, evolutionary sciences, or medical sciences. This research allows us to infer several aspects of the way of life in past populations and ancient societies. Paleopathologies, hygiene, food habits, or waste management are among topics that can be addressed by paleoparasitology. In a broader perspective, it allows to better understand the history of parasites and the evolution of the host-pathogen relationships. The intrinsic diversity of this discipline is also well illustrated by its materials and methods, as it includes respectively fossil and subfossil body parts, coprolites and paleofeces, sediments, microscopy, immunology, or molecular biology. Finally, paleoparasitology itself, but also as a result of its history as a scientific field, is closely tied to other disciplines interested in diseases throughout time, their signs and their causative agents, and complement each other, such as paleopathology, paleomicrobiology, or archaeoentomology [1–5].

The foundation of paleoparasitology can be traced back to 1910 and the publication by M.A Ruffer of the observation of *Schistosoma haematobium* eggs in ancient Egyptian mummies dated to the 20th dynasty (1250–1000 BCE) [6]. Subsequently, more than thirty years went by before the second paper with a "paleoparasitological result" appeared, with the work of Lothar Szidat [7]. He retrieved roundworm (*Ascaris lumbricoides*) and whipworm (*Trichuris trichiura*) eggs in samples from the Dröbnitz Girl in Poland, dated from around 600 years BCE, and in samples from a bog mummy, the Karvinden Man, dated from 500 years CE. From the 1950s to the 1970s, European paleoparasitology developed

in concomitance with American paleoparasitology. The definition, scope, and limits of paleoparasitological research were finally published in 1979 by Luiz Fernando Ferreira, Adauto Araujo, and Ulysse Confalonieri [8]. Since then, several highly active laboratories have emerged, following the initial Brazilian, American, and French laboratories (namely the Oswaldo Cruz Institute, the University of Nebraska, and the University of Reims Champagne Ardenne), such as the National University of Mar del Plata in Argentina, the National University of Medicine in South Korea, the University of Cambridge in England, or the University of Bourgogne Franche-Comté in France. Finally, research in paleoparasitology is also carried out by several teams of archaeology, parasitology, or biology in Russia, China, Iran, Denmark, England, or the Czech Republic [9–14].

## 2. Parasites, Parasitism, and Paleoparasitology

Not only paleosciences but also modern parasitology, provide the theoretical framework for paleoparasitology as a scientific field. In this way, interpretations of paleoenvironmental reconstructions or parasite-human relationships depend on concepts and definitions from modern parasitology.

Parasitism is a difficult notion to define as its precise shapes are unclear. Meanwhile, a minimal and operative definition might characterize parasites as organisms living at least part of their lives in or on another organism. The former (the parasite) depends upon the latter (the host) to maintain its life cycle and the beneficial relationship for the parasite is at least periodically detrimental to the host. This definition ties together those of Leuckart [15] and Brumpt [16]. According to this definition, many organisms appear to be parasitic. Price [17] estimated that 70% of all living species were parasites, while Timm and Clauson's estimation [18] was 50%. If we follow Poulin and Morand [19] and consider that each living species hosts at least one specific parasitic species, then we must indeed admit that at least half of all living species are parasites [20]. And we do not know of any living species free of parasites. Consequently, parasitism, which is more an ecological concept than a taxonomic one, emerged in many different phyla through time [21]. Several of them may even consist mostly of parasitic species, such as Apicomplexa or Acanthocephala. On the contrary, other groups never gave birth to parasitic species, e.g., primates. Meanwhile, all species are confronted with parasites in one way or another [22]. *Homo sapiens* for instance are known to be potentially infected by 179 eukaryotic parasites, including 35 specific ones [23]. Finally, parasites can be characterized by the diversity of their life cycles. Some are aquatic, others are terrestrial. Some require several hosts, be they animal or vegetal, whereas others only need one host. Some need to be involuntarily ingested through the consumption of food or spoiled hands; others actively penetrate their host.

For a long time, specialists in paleoparasitology were mainly interested in worms and unicellular parasites of the digestive tract of humans and other animals. The identification of parasites in ancient contexts is possible through the observation of preserved markers that can be classified into three categories: macroremains, dissemination and reproduction forms, and biomolecules [3].

In the case of gastrointestinal parasites, macroremains correspond to the body parts of adult worms or larvae. These macroremains, like other soft tissues, undergo rapid degradation and their preservation requires extreme dry, cold, or humid environments, or conditions favoring mineralization. Observations of digestive parasite macroremains are rare, and only five cases are known today. The two most recent are those reported by Dentzien-Dias et al. [24], who observed a body part of a tapeworm in 270-million-year-old shark coprolites, and the case reported by Charlier et al. [25], who observed a possible schistosome fragment in mummified St. Louis viscera under the electron microscope.

The second category of markers includes eggs produced by parasitic worms (or helminths) during reproduction, as well as cysts of parasitic unicellular organisms. Helminth eggs are ovoid elements whose size varies between 30 and 160 µm long and 15 to 100 µm wide. The shell of these eggs contains chitin (a polysaccharide), keratin (a fibrous protein), and sclerotin (a tanned protein), alone or associated with each other. These molecules give

them great resistance to taphonomic processes. Indeed, several parasites are oviparous helminth species, capable of producing thousands of microscopic eggs per day and female, each host being possibly infected by hundreds of worms. For these reasons, the identification of helminth eggs still represents the major activity of research in paleoparasitology. Helminth eggs are identified based on morphological criteria (shape, the presence or not of an operculum, or the ornamentation sometimes present on the shell), and their dimensions (essentially length and width). The identification of eggs is often limited to the genus, except for a few cases with characteristics allowing for identification to the species level. Protozoan cysts are spherical to ovoid in shape and vary in size from 2 to 90 μm. They are much more fragile than helminth eggs and require extreme and constant conditions of humidity, dryness, or freezing to preserve themselves. Examples of preserved cysts in archaeological contexts are very rare and the detection of protozoa requires the use of other techniques, explained below.

Among the parasite-specific biomolecules that have already been identified in archaeological material, antigens and nucleic acids (ancient DNA) are the most represented. Parasites possess a set of antigens, some of which are specific, that can be detected by adapted tools. In the last ten years, immunology has been used to detect paleoantigens of parasitic protozoa, thus increasing the list of detectable parasites in ancient contexts. As the forms of dissemination of protozoa are fragile, immunology is an effective tool for the detection of these parasites. Moreover, the specificity of certain antigenic markers allows for the direct identification of parasites, as well as their host, and consequently the characterization of the biological origin of a sample, which is a recurrent question in paleoparasitological studies. The intestinal protozoa studied so far by immunology are present-day parasites of medical importance such as *Entamoeba histolytica*, the human pathogenic amoeba, *Giardia intestinalis*, or *Cryptosporidium parvum* [26–30]. The search for ancient DNA molecules has been refined since the first publication in this field in 1984, to adapt current methods to highly degraded and diluted molecules. The first publications regarding the ancient DNA (aDNA) of parasites appeared in 2001 [31,32], and one of the first Ph.D. theses entirely dedicated to this subject was presented in 2015 [33]. Since then, the number of publications in molecular paleoparasitology has gradually risen and is now a totally integrated and state-of-the-art facet of the broader field of paleogenetics. Molecular paleoparasitology has strong advantages over previously cited methods, primarily the possibility to detect species that do not produce many eggs, but also the possibility to detect different strains of the same species and to study their phylogenetic interrelationships and molecular evolution from the past to the present. Meanwhile, the current state of the art tends to show that no method alone is sufficient to obtain a global perspective, and partially overlapping results indicate that integrative approaches should be encouraged [34]. This conclusion also applies to other fields of paleoecology [35,36].

### 3. From the Field to the Lab

The sampling strategy in the field is simple and samples are taken by the archaeologist or the anthropologist during the excavation. All structures and areas where organic remains (particularly fecal matter and intestinal residues) tend to accumulate can be targeted. These structures can be divided into two categories: 1/Human and other animal skeletons and mummies. 2/Accumulation structures, i.e., latrines, pits, sewers, or soils with accumulated organic matter. For skeleton sampling, the main samples should be taken around the pelvic region of the body, precisely just under the hip bones, the last vertebrae, and the femurs. Control samples are taken under the skull and/or the feet (Figure 1A). For accumulation structures such as latrines or pits, the sampling strategy depends on the type of filling. In case of a homogenous filling, samples should be taken regularly from the top to the bottom with a minimum of three samples. For fillings showing SU, samples should be taken from each SU (Figure 1B). In these two cases, control samples should be taken in contemporary layers outside the structure. Sediment core sampling is also possible. These approaches are particularly suitable for questions relating to variations in the function of



the structure throughout time. In the case of soil sampling, for example in occupation layers, a horizontal systematic sampling strategy can be chosen (one sample per square meter for example), or a sampling strategy covering a large part of the layer in the excavated zone, as shown in Figure 1C. This sampling strategy is obviously recommended when preservation conditions are favorable to a spatial approach and it is optimal for a synchronic diagnosis of the parasitic burden. Each sample should consist of roughly 20–100 g of sediment so that several tests can be made. Finally, for coprolite finds, possibly associated with skeletons or mixed with the fillings of accumulation structures (latrines or soils), sampling is recommended over 2–3 cm, in addition to the sediment situated just under the coprolite.

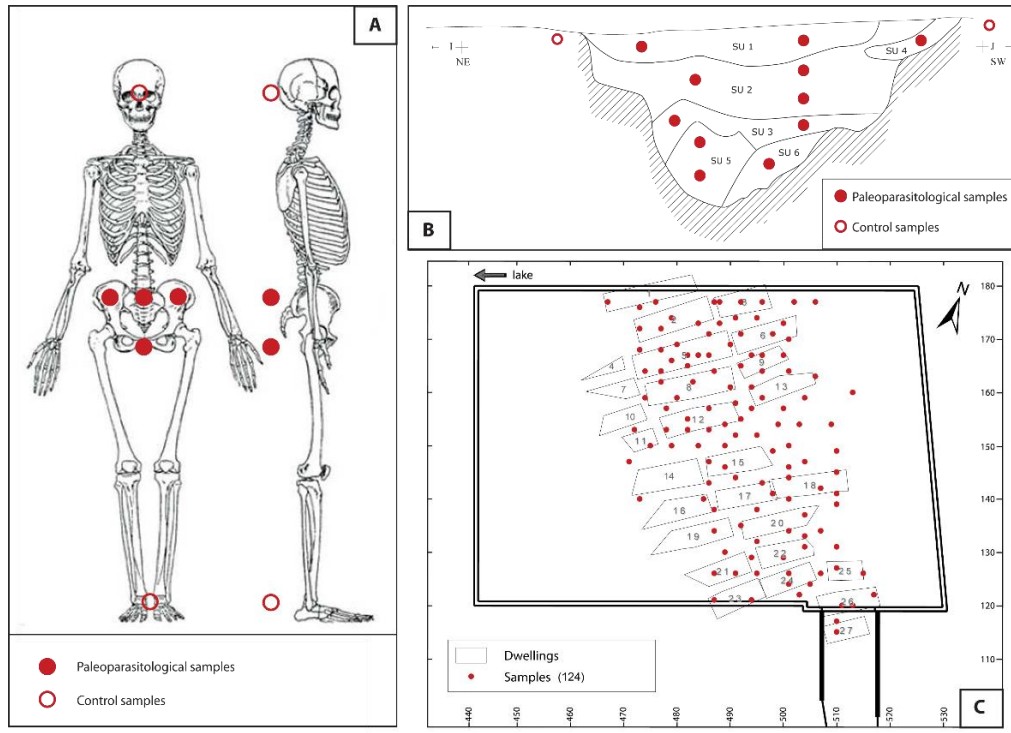

**Figure 1.** Various sampling strategies depending on various archaeological context: (**A**) sampling on skeletons; (**B**) sampling in hollowed structures as latrines or pits; (**C**) sampling in occupation layers (example of Zurich Parkhaus Opera, Switzerland).

For all types of analysis (microscopy, immunology, aDNA), clean tools and containers have to be used for each sample to avoid contamination, false-positive signals, and loss of material. In this aim, double containers are recommended. Moreover, it is preferable to collect samples as soon as possible to avoid modern contamination (by wandering animals at night, for example). Samples should be stored in dry and dark conditions, and if possible refrigerated, and no operations (e.g., sieving, sunlight drying, or chemical exposure) should be conducted prior to paleoparasitological analysis. Samples are then processed according to a specific protocol allowing for the extraction of the sought-after ancient parasitic markers. For gastrointestinal worm eggs, modern coprology methods are often of limited effectiveness due to the physicochemical changes related to taphonomic phenomena. Units specialized in the research of ancient parasites mostly use tripartite extraction techniques (rehydration, homogenization, and filtration/micro-screening) with specific adaptations [37–39]. Observations and counts are then carried out under the microscope. The study of biomolecules often requires the establishment of collaborations with specialized laboratories or takes place in specialized and adapted rooms. Examples of paleogenetic analyses are presented below.

## 4. Recent Applications of Paleoparasitology

The study of ancient gastrointestinal parasites can answer or contribute to various questions and problematics in archaeology, such as paleopathology, diet, the function of archaeological structures, waste management, or human/animal interactions. Paleoparasitology also contributes to improving our knowledge of parasite history and the evolution of host/parasite relationships, which are current topics in modern epidemiology and parasitology research. Some of these contributions are illustrated below using various examples from recent research by the paleoparasitology group of the Chrono-environment laboratory (Besançon, France).

### 4.1. Multi-Scale Analysis in Roman Sites

Paleoparasitology data analyses enable us to obtain information at different scales, ranging from the structure to the entire site, to acquire broader spatio-temporal information, for example over a defined period.

At the scale of the structure, the analysis of the filling indicates a biological origin of the filling and a function for the studied structure itself. At the site level, the location and function of several structures can provide information on waste management at the site and may highlight specific sectorizations for example. These various scales were tested on the Roman site of Horbourg-Wihr (Alsace region, France) [40]. This site corresponds to a vicus (a village in a rural area) occupied between the first and the third centuries CE. Sediment samples taken from twenty-six hollow structures of different types (pits, latrines, wells, ditches...) were studied. The results obtained for this site revealed one of the highest numbers of taxa for the Roman period (thirteen different taxa), with different and sometimes quite high numbers of eggs per structure. The identified taxa, the precise quantification of eggs per taxon, the total number of eggs, as well as the potential biological origin of the waste point to the function of the structures studied in paleoparasitology, most often latrines (i.e., only human waste) or rubbish pits (i.e., animal or mixed human/animal waste). Using the hierarchical cluster analysis (HCA), the structures were then grouped according to their filling, thus refining the proposed functions depending on the groups to which they belonged (Figure 2).

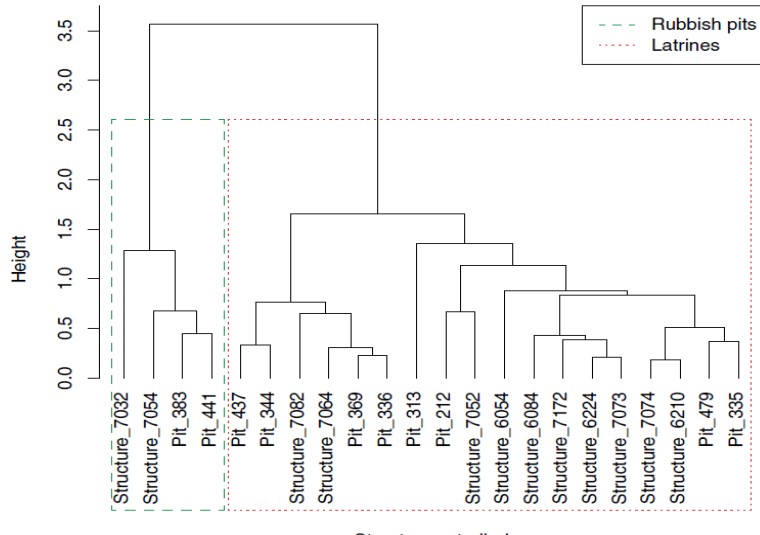

**Figure 2.** Example of hierarchical cluster analysis of structures from the site of Horbourg-Wihr. The group formed by rubbish pits is delimited by a green dashed line and that of the latrines by a red dotted line.

The results provide more general information when they are associated with the location of structures on the site. Thus, the total number of eggs per structure highlights

areas with fewer eggs. Organic waste could be less present there, which could indicate differential waste management in this area compared to the rest of the site. Moreover, this area also includes a structure containing a very large number of *Fasciola* sp. eggs (a parasite of herbivores), interpreted as a possible manure pit, and another containing *Oxyuris equi* eggs (a specific equine parasite), to be linked to a stable or a horse-related activity on the site.

The review of paleoparasitological data concerning for example a specific chronological period or geographic area provides a view of parasitic diseases at a broader supra-regional level. The review of data from the Roman period (27 BCE–476 CE) shows a significant and almost systematic presence of *Ascaris* sp. and *Trichuris* sp. in all the studied sites [40]. These two parasites have a specific direct fecal-oral transmission life cycle and are related to poor hygiene conditions. This result is even more surprising in view of the hygienic measures implemented by the Romans. Indeed, the use of public baths and latrines became widespread alongside the establishment of efficient water delivery systems (aqueducts, water pipes...) and wastewater disposal (sewers...). All these measures were therefore apparently insufficient to counterbalance public health problems caused by the growth of cities and the increase in population densities, as well as by the management of organic waste [40,41]. Public and personal hygiene at that time were indeed not comparable to current health standards. For example, public baths were open to everyone, to the sick as well as to the healthy, and there was no sanitary control at the entrance (no individual shower beforehand...) [42]. In the same manner, the Romans ate with their fingers, and handwashing, although documented at the time, was apparently not systematic and not always conscientious.

In addition, the use of waste of human and animal origin as a soil amendment for vegetable crops and private gardens (manure input and herds left to graze on fields), as well as the inadvertent consumption of food and water contaminated by such waste, also contribute to maintaining the presence of these geohelminths in Roman populations. Moreover, drinking water could also be polluted by organic waste, when wells were located near latrines, for example at the site of Horbourg-Wihr [40,41,43].

*4.2. Contribution of Paleogenetics to Paleoparasitology*

Since the first paleoparasitology-turned-paleogenetics studies in 2001 [31,32], few papers deal with parasite aDNA compared to other fields where paleogenetics is commonly applied. In 2018, Côté and Le Bailly recorded around twenty papers of this kind. Since then, a couple more studies have been published, showing new interest in molecular paleoparasitology among researchers from a variety of fields [44]. Oddly enough, the search for parasite aDNA has not been extensively performed in funerary contexts despite the high number of bioanthropological assemblages commonly discovered and studied using a wide diversity of approaches (examples include [45]; more recently [46]). In this aim, we recently studied sediment samples collected from several burials in the necropolis from the late Roman period under the basement of the Uffizi Gallery in Florence, Italy [47,48]. This funerary area was in use over a short period of time (possibly a few weeks), undoubtedly associated with an epidemic event of still unknown origin. Indeed, the corpses tended to be gathered in common pits outside the city walls and taphonomic observations indicate that this area was not used for any kind of activity before the necropolis was set up, nor in the following centuries. Sediment samples were collected from the abdominal cavity of 18 out of the 75 individuals discovered up until now to look for intestinal parasites. The RHM protocol [37] for helminth egg extraction was applied, and five individuals showed Ascarid-type specimens, most probably related to *Ascaris lumbricoides*, a human-infesting parasite associated with fecal pollution (Figure 3A). Following this microscopy-based diagnosis, 10 sub-samples from five individuals (three of which were previously positive for Ascarid-type eggs under light microscopy) were tested for the ancient DNA of a diversity of intestinal parasites. The whole-sediment-DNA silica-based extraction was followed by targeted PCRs, then cloning and sequencing positive signals. Paleogenetics greatly

expanded previously obtained results as all five individuals tested positive for at least one aDNA parasite, confirming the presence of *Ascaris* sp. where Ascarid-type eggs were previously recorded. Moreover, two more taxa appeared as five individuals tested positive for *Trichuris trichiura*, and one individual tested positive for *Dicrocoelium dendriticum*.

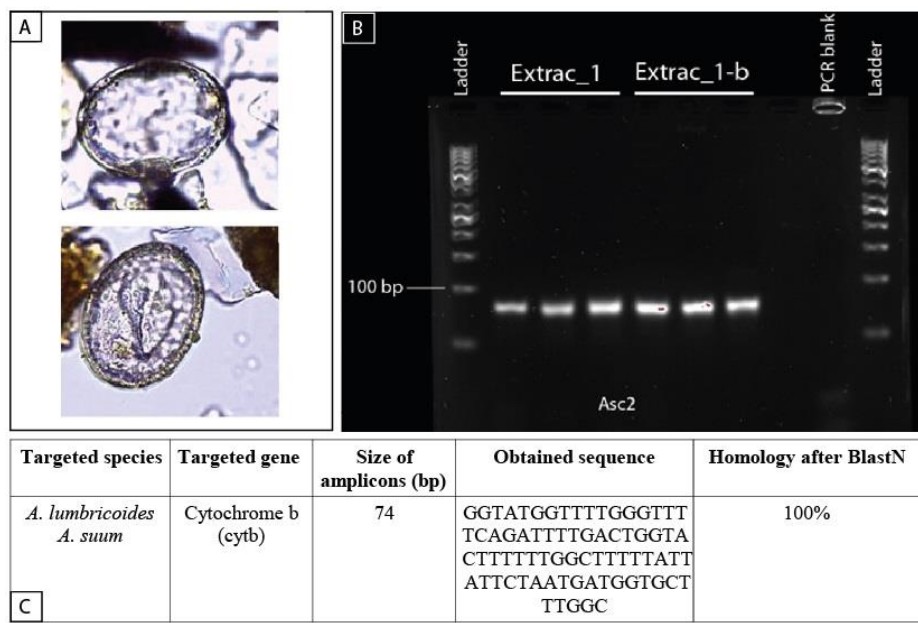

**Figure 3.** (**A**) Ascarid-type eggs corresponding to decorticated "*Ascaris* eggs" observed under the Uffizi Gallery in Florence, 62 μm × 47 μm each, 400×. (**B**) Visualization on electrophoresis gel (2%) of obtained amplicons from 6 PCRs reactions in 2 sediment DNA extractions showing a clear positive reaction for the targeted taxon (*Ascaris* sp.). (**C**) Targeted gene for *Ascaris* sp. DNA amplification and obtained sequence. The 74 bp long fragment showed 100% homology with the currently known *Ascaris* sp. genome when blasted in Genbank.

Even if the latter case may be called into question due to possible DNA leaching, this integrative approach allowed us to recognize potential samples under the microscope before testing them for aDNA, and extended previously known parasite diversity and diffusion in this population sub-group. These observations are in line with previous conclusions in paleoparasitology and other fields of paleoecology, regarding the usefulness of integrative approaches combining classical and molecular proxies for a more comprehensive picture of past environments. Moreover, it contributes, with previous papers, to pave the way for a finer comprehension of past infectious dynamics in human populations through time and a better understanding of endemic and epidemic disease interrelationships.

### 4.3. Using Statistical Treatment and Spatializing Paleoparasitological Data

As mentioned above, a comprehensive analysis of an archaeological site with a complete parasitological diagnosis is achievable. By sampling a specific layer, with well-defined dates, it is possible to spatialize areas of parasite marker concentration. Materializing concentrations of parasite eggs, apprehending specific areas of fecal matter discharge which can be linked to latrines or stable areas, for instance, becomes conceivable. This approach has been established on several Neolithic lake sites, the site of La Draga (Spain) and Zurich-Parkhaus Opéra (ZPO, Switzerland). It should be noted that exceptional preservation conditions are necessary to undertake this type of approach.

The site of La Draga constitutes the first evidence of a Neolithic lake settlement for the Iberian Peninsula and provides an opportunity to explore site occupation as well as the economic exploitation of the surrounding resources [49,50]. A total of 72 samples were taken from the oldest layer dated from the second half of the sixth millennium BCE. Parasitic markers point to a high concentration area in a dwelling, indicating a specific zone

of domestic and fecal waste matter (Figure 4A). This hot spot demonstrates a tendency to manage waste at the household level. Despite their willingness to concentrate waste, the inhabitants present a high rate of infestation and show co-occurrence of several gastro-intestinal parasites [51,52].

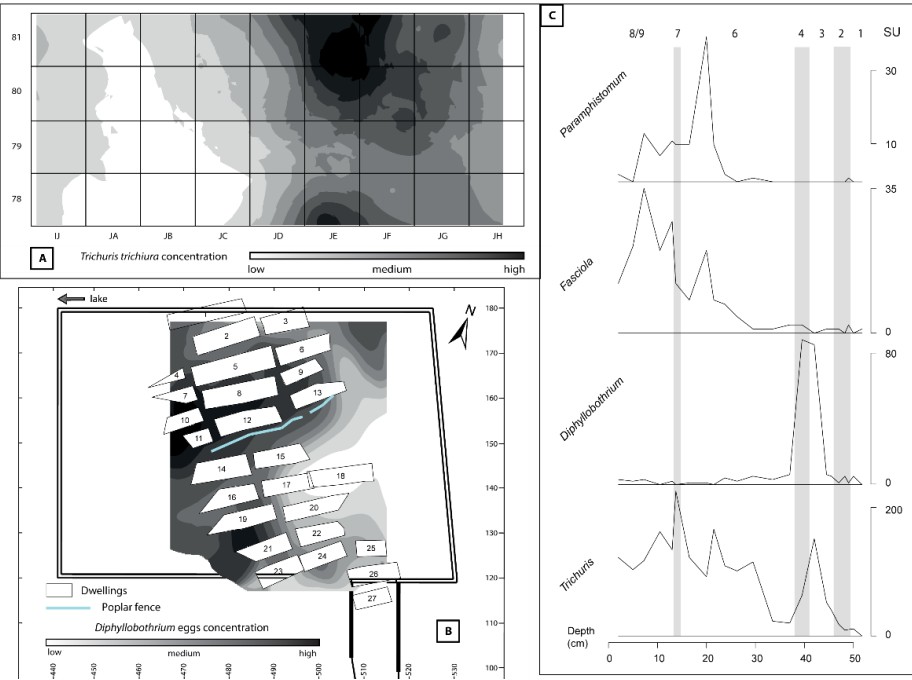

**Figure 4.** Mapping parasitological residues allow showing concentration areas corresponding to particular rejection zones in archaeological sites. Example in La Draga, Spain (**A**) and Zurich Parkhaus Opera, Switzerland (**B**). (**C**) Evolution of the parasite spectrum in Zug-Riedmatt, Switzerland.

More recently, a larger-scale study was carried out on the site of Zurich Parkhaus Opera, with a set of 124 samples analyzed from the only layer of the site, dating from 3176 to 3153 BCE. The possible dissociation between the domestic context and circulation paths for data processing revealed a set of information. Firstly, a cattle stable area is evidenced by high concentrations of ruminant parasites. However, most importantly, an as of yet unexplored perspective for paleoparasitology emerged from this study. In fact, this site is singular, not only on account of exceptional preservation conditions but because of the identification of population sectorization, materialized by a physical barrier (poplar barrier) [53,54]. This observation is reflected in the parasite spectrum identified over the whole site, which differs according to the social status of inhabitants (Figure 4B). These specific synchronic contexts provide valuable information about the sectorization of the population and the resulting waste management [50].

A good example of a diachronic analysis is the Neolithic site of Zug-Riedmatt (Switzerland), dated to 3250 to 3100 cal. BCE [55]. At this site, a set of core samples, and not a single occupation layer, was sampled and analyzed. Micromorphology and radiocarbon dating point to two or three occupation phases, without interruptions [56,57]. The evolution of the parasitic spectrum demonstrates a drastic change in livelihood patterns over time, ranging from a strong dominance of fish consumption to an increase in meat products (Figure 4C). However, the question of the existence of two or three occupation phases remains unresolved. Although a change in supply is perceptible through the parasitic spectrum, it is difficult to prove that this evolution reflects distinct occupation phases or even a change in environmental conditions. In short, the knowledge provided by paleoparasitology varies according to the observation scale, ranging from the state of health of the population to waste treatment, including the management of food resources.

*4.4. Integration of Paleoparasitology in Paleoecological Reconstructions*

We recently integrated a systematic paleoparasitological investigation in a multi-proxy peat bog core study. The peat bog of Asi Gonia is located at an altitude of 750 m in the White Mountains, Western Crete, Greece. A 6 m core was taken from the deepest part of the bog, and 15 radiocarbon samples were taken, dating the beginning of the site to the first century BCE. Among others, a very important accumulation of fungal spores was observed, particularly marked during the Roman period [58].

Coprophilous and saprophytic fungi are traditionally recognized in paleoecological studies as reliable markers for the presence of herbivorous mammals in past environments, and particularly of rising pastoral pressure in terrestrial ecosystems during the Holocene [59–62]. Sporadic observations of parasite eggs in palynological studies incited previous researchers to propose the systematization of paleoparasitological research alongside other biomarkers in reconstructions of past environments [63]. However, such systematic studies have not really materialized, either in peat bog core studies or by following the standards of this discipline to avoid egg degradation due to very stringent palynological methods.

Twenty-two sediment samples were collected from this peat bog core. We aimed to test the feasibility of paleoparasitological studies in this kind of context, and, if applicable, to trace animal presence and pastoral pressure in the watershed of Asi Gonia in conjunction with the analysis of pollen grains and coprophilous fungal spores [64]. A total of 19 samples (86%) tested positive for parasite eggs under light microscopy. Despite the relatively small number of eggs (58 observed eggs in the whole sequence), they were particularly well preserved, ranging from the Roman period to the twentieth century, and represented at least nine morphotypes corresponding to five Nematoda genera, two Trematoda genera, and one Acanthocephala genus, plus a currently unknown Trichocephalida morphotype.

In conjunction with fungal spore and pollen grain records, these results suggest seemingly coeval dynamics for all our biomarkers, showing notably a higher intensity and diversity of coprophilous fungal spores alongside a higher diversity of parasite taxa per sample during a first period ranging from the Roman period in the first c. BCE to 380 CE in an evergreen oak forest landscape. Regarding the zoo-archaeozoological record in Crete, this was tentatively interpreted as the sign of acorn-fed pig-herding in the oak-forested watershed at that time. Interestingly, this pastoral activity, which is still practiced in the Mediterranean, had not been previously suggested for ancient Crete. After this first period, parasitological and fungal remains follow their coeval dynamics through a marked decline in diversity and intensity from 380 to 900 CE. This period also appears to be an ecological tipping point from the former oak forest towards a heather maquis landscape. From 900 to 1850 CE, a renewed and regular rise in coprophilous fungal spores occurred, although their numbers did not attain previous levels of intensity and diversity. Again, parasite eggs appear regularly throughout this period, alongside the disappearance of Acanthocephala remains (typically associated with pig parasites), and the very occasional appearance of ruminant parasites (*Paramphistomum* sp. and *Fasciola* sp.). This may be the sign of a decrease in pastoral pressure in conjunction with a change in animal herding towards a more ruminant-based practice in this open environment. The last phase, starting in the mid-nineteenth century is again marked by a sharp decrease in parasite and fungal spore counts, and also corresponds to a new ecological tipping point from a heather maquis toward phrygana steppes. While the decline observed at the beginning of the period may be related to the historically known abandonment of herding activities in the region at that time, it is interesting to see how fungal spores rise very strongly again in the second half of the twentieth century. This rise is undoubtedly associated with European funding for the re-introduction of herding activities at that time. Interestingly, this new rise in fungal spore accumulation corresponds to those observed in the Roman period, suggesting very important pastoral activity in the watershed at that time. In the meantime, hardly any parasite eggs were detected for this period, suggesting the introduction of veterinary treatments against helminthiasis at that time, confirmed

by the analysis of surface soil and fresh feces samples in the coring area, that remained negative after coprological examination.

Regarding the low number of observed eggs, the previously exposed interpretations in this precise case must be taken with caution. Meanwhile, this pilot study showed a remarkable state of preservation of a high diversity of helminth eggs in a 2000-year-old peat bog core, enabling interpretations in conjunction with well-documented biomarkers in a well-dated sequence. The use of a standard protocol for paleoparasitological analysis probably facilitated the extraction of eggs that went unnoticed during the palynological study. In sum, this study calls for the systematic integration of paleoparasitological analysis following the standards of this discipline in multi-proxy analyses of natural archives, particularly in peat bogs.

*4.5. Contribution to Our Knowledge of Parasite History*

The multiplication of paleoparasitological studies for different periods and in different geographical areas fuels, in a second stage, questions related to the history of parasitic diseases. As a result of the compilation of data collected during analyses, it becomes possible to monitor parasites over time (appearance, disappearance, resurgence, migration), and to place these phenomena in parallel with the history of human or animal populations. In 2010, the synthesis of available data on the small liver fluke, *Dicrocoelium dendriticum*, a parasite of herbivores, highlighted the role of modern European migrations in the transfer of the parasite from Europe to the American continent [65]. This same work on the horse pinworm (*Oxyuris equi*) highlighted an increased presence of the parasite and a widening of its range in Roman-era sites, probably because of the use of the horse at that time for trade, conquests, or population movements [66] (Figure 5).

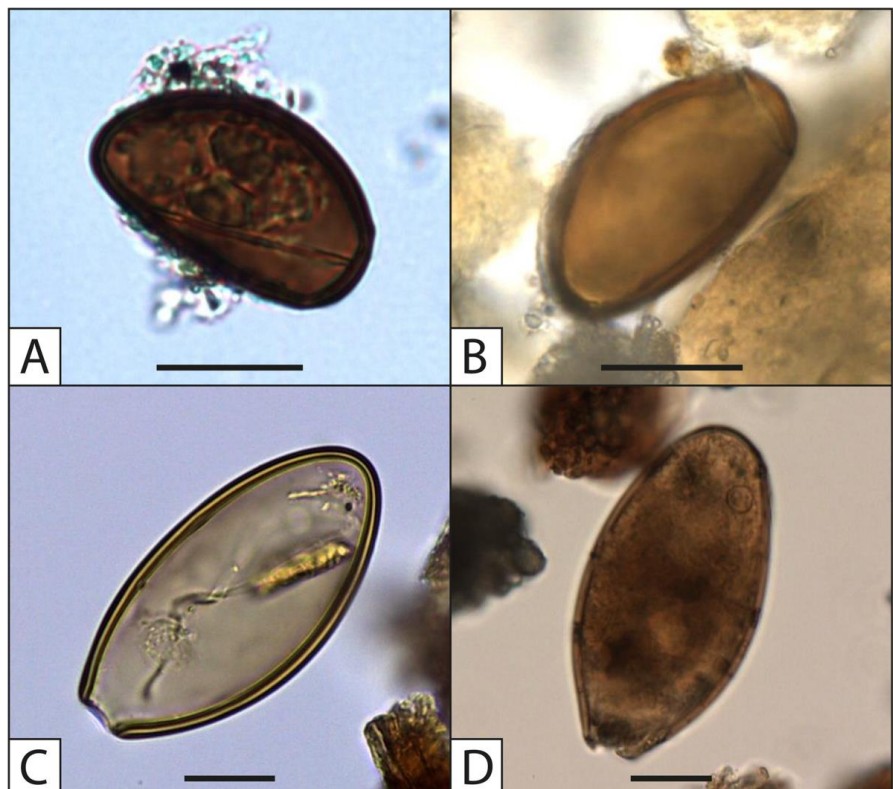

**Figure 5.** Eggs from the oldest and most recent period observed in the Besançon laboratory: (**A**) Lancet liver fluke egg from Concise (Neolithic period, Switzerland), 41.48 μm × 24.59 μm; (**B**) Lancet liver fluke egg from Reims (Modern period, France), 42.78 μm × 23.12 μm; (**C**) *Oxyuris equi* egg from Berel (Iron Age, Kazakhstan), 84.28 μm × 43.12 μm; (**D**) *Oxyuris equi* egg from Bourges (Medieval period, France), 85.65 μm × 45.23 μm. Scale bars, 20 μm.

The use of immunology since the early 2000s has also provided information for some unicellular parasites of humans. For example, the human pathogenic amoeba, *Entamoeba histolytica*, present in Europe since the Middle Neolithic, seems to have appeared on the American continent around the twelfth century, three centuries before the 'discovery' of this territory by Christopher Columbus. The nature of the migratory phenomena behind this transfer remains unknown, but these analyses have challenged the old assumption of an American origin for the parasite [29]. The history of many other parasites remains to be elucidated and only further analysis will provide some answers. This is the case, for example, for the roundworm, whose almost systematic presence in paleoparasitological assemblages makes monitoring and interpretations difficult. This parasite, detected in Europe in Paleolithic samples, is predominant in parasitic fauna spectra from the Roman period onwards and remains so until the most modern periods of history [40,43].

## 5. Conclusions

After more than a century of development and data acquisition, paleoparasitology has become a recognized bioarchaeological discipline, helping to answer questions from archaeology, and developing its own issues related to the history of parasites and the evolution of host/parasite relationships. Methodological advances and the increasing use of new investigative techniques are promising, especially concerning the detection and identification of parasites, two crucial points on which scientists interested in ancient parasites must work. In this perspective, the use of molecular biology must continue and combined microscopy/immunology/molecular biology approaches must become increasingly widespread to obtain more complete images of parasite diversity and evolution at different periods of history.

The actions carried out so far through archaeological training or data dissemination must continue so that paleoparasitology develops further and occupies a more important place in the archaeological community. Specialists must continue efforts to enhance the integration of data in archaeology. These steps will perhaps make it possible to systematize paleoparasitological studies in certain contexts (latrines, burials…) in the same way as other disciplines such as archaeozoology and archaeobotany.

**Author Contributions:** M.L.B. managed the article writing and edition, and wrote Section 4.5. C.M. wrote Section 4.3. K.R. wrote Sections 4.2 and 4.4. B.D. wrote Section 4.1. All authors co-wrote the introduction, chapters 2 and 3, and conclusion. All authors have read and agreed to the published version of the manuscript.

**Funding:** This research received no external funding.

**Institutional Review Board Statement:** Not applicable.

**Informed Consent Statement:** Not applicable.

**Data Availability Statement:** Not applicable.

**Acknowledgments:** The authors would like to thank all their collaborators in the field, archaeologists, anthropologists, and archaeozoologists, and in the lab, biologists, parasitologists, and geneticists.

**Conflicts of Interest:** The authors declare no conflict of interest.

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
