# Peer review of "Accessing Ancient Population Lifeways through the Study of Gastrointestinal Parasites: Paleoparasitology"

_applsci, doi:10.3390/app11114868_

Round 1

Reviewer 1 Report

Thank you for the opportunity of reviewing this paper

The paper is extensively researched and provides a wealth of review information in a subdiscipline of paleopathology: palaeoparasitology. There is no question that the presentation of material of this nature, gathered in one place, will be useful and interesting too many readers of Applied Sciences. However, before the paper can proceed to publication, the key points need to be brought out more strongly and summarised to assist the reader.

What insights have been gained into the lifeways of ancient populations through palaeoparasitology that have not been provided by other disciplines? In what areas of understanding have the main advances been made? The answers to these questions are widely dispersed as disparate pieces of information throughout the text, and need to be brought together if they are to support the title of the paper. This deficiency is highlighted by the fact that the abstract is effectively content-free and it does not address the title: the abstract should contain a summary of the key findings and themes that palaeoparasitology has provided in relation to the lifeways of ancient peoples, as promised in the title.

If the authors can address the problem I have highlighted, the paper would be greatly strengthened and certainly worthy of publication. It would also need editorial review because the English is sometimes (understandably) idiosyncratic; for example, lifeways would be better used in the plural, and AEC is not an abbreviation we use in English.

These criticisms notwithstanding, the authors have produced a detailed review of a lot of material that is intelligently extracted from the wealth of literature available. I would therefore encourage them to persist with the paper to bring it to the requisite standard for publication.

I trust that these comments are helpful

Reviewer 2 Report

The manuscript entitled ‘Accessing ancient population lifeway by the study of their gastrointestinal parasites: paleoparasitology’ is a great contribution to this Special Issue in Applied Sciences, as provides a complete review on the theoretical framework, the history of the discipline and its methodology. Besides, it provides with case study applications that are really illustrative of the potential of paleoparasitology in archaeological and palaeoecological research. This paper is suitable for the Applied Sciences scope and will consist of a reference work on the field of paleoparasitology.

I absolutely recommend the publication of this manuscript in the current version. I only suggest one minor change:

-line 306: check the chronology of La Draga. The oldest phase is around 7200 cal BP/5200 cal BC, that is the 2nd half of the eighth/sixth millennium.

Reviewer 3 Report

A very interesting work on palaeoparasitology, especially endoparasites.

I recommend this manuscript for publication in the Applied Science after minor corrections.

The manuscript is of an review character. However, I do not understand the description of the contribution of individual authors - individual analyzes were made within the framework of other, original, probably already  published articles ?. If the work contains original data, the manuscript format must be changed - clearly separate the review part from the original, i.e. new results.

Detailed comments

1. Line 34: „… observation of S. haematobium…” – should be “… observation of Schistosoma haematobium …”.

2Line 58: “…of their life in or on another organism…” – please add “specimens” – “… of their life in or on another organism/specimens.” There is / must be another species in the definition of a parasite, not only organism.

3. Line 58: “…Price estimated” – please add citation – “Price […]”.

Line 62: „…Tim and Clauson” – the same comment,

Line 63: “…Poulin and Morand..” - the same comment; unless citation number 16 refers to it ?

4. Line 70: please use the entire generic name (Homo) at the beginning of the sentence.

5. Figure 1-5: please provide source/citation; please see general comments (Author contributions). Moreover, please indicate whether the data (figures) are original or any changes have been made.

6. Lines 366, 382: “…Acanthocephalan …” – please change – “Acanthocephala” or “acanthocephalan”.

7. Line 414: there is a "Dicrocoelium dendriticum" in the text, and a "Dicrocoelium sp." in the Figure 5 - is it supposed to be like that?

8. References inconsistent with the requirements of the Applied Science, please correct.

Round 2

Reviewer 1 Report

Can now proceed to publication